# ALDi: Quantifying the Arabic Level of Dialectness of Text

**Amr Keleg, Sharon Goldwater, Walid Magdy**

Institute for Language, Cognition and Computation

School of Informatics, University of Edinburgh

akeleg@sms.ed.ac.uk, {sgwater,wmagdy}@inf.ed.ac.uk

## Abstract

Transcribed speech and user-generated text in Arabic typically contain a mixture of Modern Standard Arabic (MSA), the standardized language taught in schools, and Dialectal Arabic (DA), used in daily communications. To handle this variation, previous work in Arabic NLP has focused on Dialect Identification (DI) on the sentence or the token level. However, DI treats the task as binary, whereas we argue that Arabic speakers perceive a spectrum of dialectness, which we operationalize at the sentence level as the *Arabic Level of Dialectness* (ALDi), a continuous linguistic variable. We introduce the AOC-ALDi dataset (derived from the AOC dataset), containing 127,835 sentences (17% from news articles and 83% from user comments on those articles) which are manually labeled with their level of dialectness. We provide a detailed analysis of AOC-ALDi and show that a model trained on it can effectively identify levels of dialectness on a range of other corpora (including dialects and genres not included in AOC-ALDi), providing a more nuanced picture than traditional DI systems. Through case studies, we illustrate how ALDi can reveal Arabic speakers' stylistic choices in different situations, a useful property for sociolinguistic analyses.

## 1 Introduction

Arabic is spoken by more than 420 million people all over the world (Bergman and Diab, 2022), and exists in a state of *Diglossia*, in which two variants of the language co-exist in Arabic-speaking communities (Ferguson, 1959). Modern Standard Arabic (MSA) is the standardized variant, which is taught in schools and used in formal communications and as a common language across all Arab countries. However, many local variants of Dialectal Arabic (DA) are used for daily communication—mainly in speech and speech-like text such as social media. They can diverge from MSA and each other in phonology, morphology, syntax, and semantics

| Level of Dialectness | Egyptian | Levantine |
|---|---|---|
| **MSA** | أسعدنا الرجل | أسعدنا الرجل |
| **Low** | الراجل أسعدنا | الزلمة أسعدنا |
| **Medium** | الراجل بسطنا | الزلمة بسطنا |
| **High** | الراجل شهيصنا | الزلمة نغنجنا |

Table 1: Example sentence meaning *the man cheered us* written with different levels of dialectness in two Arabic dialects. Words with DA features are underlined. The dialectal sentences use their preferred SVO word order, contrasted by VOS order for MSA. The low dialectness example also shows a lexical dialectal feature for the word *the man* (MSA الرجل): the Egyptian word (الراجل) differs from MSA in a single character, while the equivalent Levantine word (الزلمة) has a different origin. Both dialects allow different variants for the verb: one variant (بسطنا), used in both dialects, shares a root with the MSA variant, while the more dialectal variants (شهيصنا in Egyptian and نغنجنا in Levantine) do not.

(Habash, 2010)—sometimes even being mutually unintelligible (Abu Farha and Magdy, 2022)—and they do not have a standard orthography.

These differences between MSA and DA, and the fact that speakers commonly code-switch between the two, are a major challenge for Arabic NLP systems. As a result, many systems have been designed to perform Dialect Identification (DI), often on the sentence level (Zaidan and Callison-Burch, 2011; Elfardy and Diab, 2013; Salameh et al., 2018), but also on the token level as a way of detecting code-switching points (Solorio et al., 2014; Molina et al., 2016). Both formulations take a binary view of the problem (a sentence or token is either MSA or DA), and assume all the features of DA have the same impact on the perceived "dialectness" of a sentence. We argue, however, that the level of dialectness of a sentence is a spectrum, as illustrated in Table 1. Earlier initiatives recognized the presence of such a spectrum (Habash et al., 2008; Zaidan and Callison-Burch, 2011), however,

the datasets that were developed are either skewed toward more standardized documents with limited code-switching or lack information about the distribution and the quality of the levels of dialectness labels. Consequently, the *Level of Dialectness* has not yet been adopted as a linguistic variable that is formally recognized in analyzing Arabic text, despite being potentially useful for NLP applications.

We argue that the level of dialectness is an important but overlooked aspect of Arabic text which is complementary to, and more nuanced than, dialect identification. To support this claim and promote further research in the area, we:

1. Define the ***Arabic Level of Dialectness (ALDi)*** as a continuous linguistic variable that quantifies the dialectness of a sentence (or sentence-like unit) and can enrich the analysis of Arabic text.

2. Release *AOC-ALDi*[1], a dataset of 127,835 Arabic comments with their ALDi labels, which is derived from the Arabic Online Commentary dataset (Zaidan and Callison-Burch, 2011). We provide the first detailed analysis of the *level of dialectness* labels and form canonical splits for the *AOC-ALDi* dataset.

3. Propose an effective method for estimating the ALDi of sentences, that can generalize to corpora of other genres and dialects [2].

4. Demonstrate via case studies that ALDi estimation of transcribed political speeches can highlight interesting insights that existing DI systems fail to detect.

We hope that our work on the *Level of Dialectness* variable can motivate research in this direction applied to other languages such as Swiss German where a Standard variant co-exists with non-standardized ones.

## 2 Background and Related Work

**MSA and Dialectal Arabic**  Unlike English, where there is no single standard variant used in all English-dominant countries, Arabic speakers agree to a great extent on having a single standardized form of the language that they call *Fus-ha* فصحى. Arabs perceive both MSA and Classical Arabic (CA), the variant of Arabic that dates back to the 7th century, as *Fus-ha* (Parkinson, 1991).

While Arabs can understand and read this standard language, spontaneously speaking in the standard language is not a natural task for most of them. Variants of DA are generally used in everyday communications, especially in spontaneous situations, and are widely used on social media platforms.

DA variants can be grouped on the level of regions (5 major variants: Nile Basin, Gulf, Levant, Maghreb, and Gulf of Aden), countries (more than 20 variants), or cities (100+ variants) (Baimukan et al., 2022). In text, MSA differs from DA in terms of morphemes, syntax, and orthography. These differences form **cues of dialectness** in code-switched text. In the orthography, distinctive DA terms are written in ways that match the pronunciation. Regional differences in the pronunciation of MSA terms are typically lost in writing due to the standardized orthography, but in some cases, individuals use non-standard orthography that matches their regional pronunciations (e.g.: *Man* written as راجل instead of the standardized form رجل as in Table 1).

**Arabic Dialect Identification**  Due to the rise of social media text, handling DA has become increasingly important for Arabic NLP systems. To date, researchers have focused on Dialect Identification (DI), which can be modeled either as a binary MSA-DA classification or a multi-class problem with a prespecified set of DA variants (Althobaiti, 2020; Keleg and Magdy, 2023). Arabic DI has attracted considerable research attention, with multiple shared tasks (Zampieri et al. 2014; Bouamor et al. 2019; Abdul-Mageed et al. 2020, 2021b, 2022) and datasets (Zaidan and Callison-Burch, 2011; Bouamor et al., 2014; Salama et al., 2014; Bouamor et al., 2018; Alsarsour et al., 2018; Zaghouani and Charfi, 2018; El-Haj, 2020; Abdelali et al., 2021; Althobaiti, 2022).

Much of this work has been done at the sentence or document level, but there has also been work on token-level DI for code-switching, for example on Egyptian Arabic-MSA tweets (Solorio et al., 2014; Molina et al., 2016) and on Algerian Arabic (Adouane and Dobnik, 2017).

**Levels of Dialectness**  Both sentence-level and token-level DI methods fail to distinguish between sentences having the same number of dialectal cues, yet different levels of dialectness. As per Table 1, each of the sentences الزلمة بسطنا and الزلمة نغنجنا has two lexical cues of dialectness, yet the

---
[1]The code and data files can be accessed through `https://github.com/AMR-KELEG/ALDi`
[2]A live demo for ALDi estimation: `https://huggingface.co/spaces/AMR-KELEG/ALDi`

latter sentence is perceived as being more dialectal than the former. Only a very few works have considered this distinction. One is Zaidan and Callison-Burch (2011), who collected sentence-level dialectness annotations in the Arabic Online Commentary data set. Although the dataset has been released, there has been no published description or analysis of these annotations that we know of, and (perhaps for this reason) no follow-up work using them[3]. Our work aims to remedy this.

An earlier project that annotated dialectness was Habash et al. (2008), who proposed a word-level annotation scheme consisting of four levels: (1) Pure MSA, (2) MSA with non-standard orthography, (3) MSA with dialect morphology, and (4) Dialectal lexeme. Annotators also labeled full sentences according to their level of dialectness. Although the inter-annotator agreement was relatively good (less so for the sentence level), only a small corpus was annotated (19k words). Moreover, the corpus has sentences that are mostly in MSA with limited code-switching. A later work piloted a simplified version of the scheme on another corpus of 30k words (Elfardy and Diab, 2012). Both corpora are not publicly released.

**Level of Dialectness and Formality** Formality is a concept that has been studied, yet it does not generally have an agreed-upon definition (Heylighen and Dewaele, 1999; Lahiri, 2016; Pavlick and Tetreault, 2016; Rao and Tetreault, 2018). Heylighen and Dewaele (1999) define formality as the avoidance of ambiguity by minimizing the context-dependence, and the fuzziness of the used expressions. Later operationalizations recognize factors such as slang words and grammatical inaccuracies have on the people's perception of formality (Mosquera and Moreda, 2012; Peterson et al., 2011) as cited in (Pavlick and Tetreault, 2016).

Arabic speakers tend to use MSA in formal situations, and their regional dialects in informal ones. However, an Arabic speaker can still use MSA and speak informally, or use their dialect and speak formally. The case studies described in §5 show how Arab presidents use sentences of different levels of dialectness in their political speeches. While these speeches would all be considered to be formal, using different levels of dialectness might be to sound authoritative (using MSA) or seek sympathy (us-

---

[3]This contrasts with the authors' DI annotations for the same corpus, which were analyzed in Zaidan and Callison-Burch (2014), and have been widely used in DI tasks.

| Type | AlGhad | AlRiyadh | Youm7 |
|---|---|---|---|
| Comment | 94,236 | 156,345 | 80,349 |
| Control | 48,210 | 8,925 | 9,051 |
| All | 142,446 | 165,270 | 89,400 |

Table 2: Statistics of the AOC dataset, showing the number of annotations of each type from each newspaper source. Each sentence has 3 independent annotations.

ing a regional dialect). Therefore, we believe the level of dialectness and formality are related yet not interchangeable.

## 3 The Arabic Level of Dialectness (ALDi)

We define the *Level of Dialectness* of a sentence as the **extent by which the sentence diverges from the standard language**, which can be based on any of the cues described above. This definition is consistent with the crowd-sourced annotation of the Arabic Online Commentary (AOC) dataset (Zaidan and Callison-Burch, 2011), where annotators labeled user comments on Arabic newspaper articles by their *dialect* and their *level of dialectness*. However, the original and subsequent work only used the dialect labels, and the dialectness annotations have not previously been analyzed in detail.

### 3.1 Analyzing the AOC Dataset

The AOC dataset was created by scraping user comments on articles from three different newspapers, which are published in Egypt (*Youm7* - اليوم السابع), Jordan (*AlGhad* - الغد), and Saudi Arabia (*AlRiyadh* - الرياض); thus expecting the majority of comments of each source to be in Egyptian (EGY), Levantine (LEV), and Gulf (GLF) dialects respectively. Each comment is labeled for its *level of dialectness* (*MSA, little, mixed, mostly dialectal, not Arabic*). For comments labeled as Non-MSA, the annotators also chose the *dialect* in which the text is written: EGY, LEV, GLF, Maghrebi (MAG), Iraqi (IRQ), General (GEN: used when the text is DA, but could belong to multiple dialects), Unfamiliar, and Other.

Each row of the released AOC dataset consists of 12 different sentences representing a Human Intelligence Task (HIT) on Amazon mTurk, with annotations provided by the same human judge. A HIT has 10 comments in addition to 2 control sentences sampled from the articles' bodies, which are expected to be mostly written in MSA. As part of each HIT, annotators provided some personal information such as their place of residence, whether

| Type | MSA | Little | Mixed | Most | Not Arabic | Missing |
|---|---|---|---|---|---|---|
| **Comment** | 189,020 (57.12%) | 36,930 (11.16%) | 21,622 (6.53%) | 76,284 (23.05%) | 5,421 (1.64%) | 1,653 (0.5%) |
| **Control** | 62,456 (94.36%) | 1,060 (1.6%) | 436 (0.66%) | 754 (1.14%) | 1,165 (1.76%) | 315 (0.48%) |
| **All** | 251,476 (63.33%) | 37,990 (9.57%) | 22,058 (5.55%) | 77,038 (19.4%) | 6,586 (1.66%) | 1,968 (0.5%) |

Table 3: The distribution of AOC's *Level of Dialectness* annotations. Each sentence has 3 independent annotations. *Control* are sentences extracted from the article body, most likely MSA, to check the quality of the annotations.

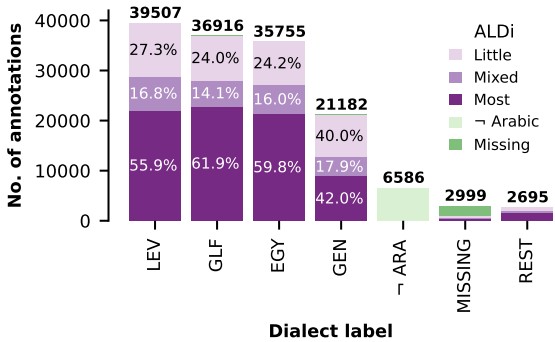

Figure 1: The distribution of the annotations for the dialect and the level of dialectness in AOC. Note that each comment has three different annotations. 251,476 MSA annotations are not shown in the Figure. The *General* dialect label is used when a sentence is natural in multiple variants of DA. The *REST* bar represents the (Maghrebi, Iraqi, Unfamiliar, and Other) labels.

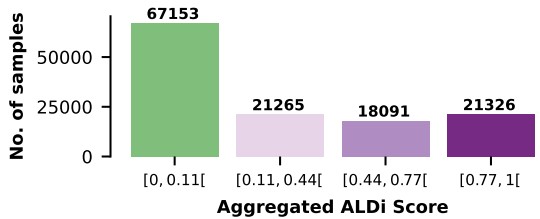

Figure 2: AOC-ALDi's distribution of ALDi scores.

they are native Arabic speakers, and the Arabic dialect they understand the most. Table 2 shows the number of annotations collected for sentences from each source.

Table 3 shows the distribution of Level of Dialectness annotations in AOC. As expected, the control sentences are nearly all (94%) annotated as MSA. MSA is also the most common label for the scraped comments (57% of their annotations), followed by the mostly dialectal label (23%), little dialectal (11%), and mixed (6.5%).

Figure 1 shows the distribution of dialectness labels split out by dialect (sentences labeled as MSA are not shown). We see that the proportions of different levels of dialectness for the LEV, GLF, and EGY dialects are similar, even though the total

number of annotations per source (Table 2) is more skewed. This is likely due to the fact (noted by Zaidan and Callison-Burch 2014) that AlGhad contains the highest proportion of MSA annotations, followed by AlRiyadh and then Youm7. Figure 1 also shows that the distribution of dialectness levels is similar for the LEV, GLF, and EGY dialects, whereas the GEN dialect label has a higher proportion of *little* dialectness. This makes sense, since for sentences with few cues of dialectness, the level of dialectness would be low, and it would be hard to assign these sentences to a specific dialect.

### 3.2 From AOC to AOC-ALDi

In order to transform the AOC level of dialectness annotations into numeric ALDi scores, we applied the following steps:

**Step #1 - HIT to annotation rows**: We split each row (HIT) of the AOC dataset into 12 annotation rows, one for each sentence of the HIT, with the annotator's information shared across them.

**Step #2 - Grouping identical comments**: Comments on the same article can sometimes be identical. We decided to group identical comments on the same article together. Out of 129,873 grouped comments, only 1,377 comments have more than three annotations. We discarded 2,038 grouped comments for which at least $\frac{2}{3}$ of the dialectness level annotations are either *Missing* or *Not Arabic*,[4] leaving a total of 127,835 comments with at least three annotations each. The average length of these comments is 20 words.

We measured inter-annotator agreement on the *level of dialectness* annotations for the 124,257 comments which have 3 annotations that are not *Not Arabic* or *Missing*. The Fleiss' Kappa ($\kappa$) is 0.44 (Fleiss, 1971), while Krippendorff's Alpha (interval method) ($\alpha$) is 0.63 (Krippendorff, 2004). Both metrics are corrected for chance agreement and disagreement respectively. $\kappa$ considers the labels as categorical, while $\alpha$ penalizes disagreements according to the differences between their val-

---

[4]The main categories of these discarded comments are discussed in Appendix §B.

| Split | AlGhad | | AlRiyadh | | Youm7 | |
|---|---|---|---|---|---|---|
| | Cmnt | Cntrl | Cmnt | Cntrl | Cmnt | Cntrl |
| **Train (80%)** | 24,039 | 12,613 | 41,479 | 2,335 | 20,041 | 2,379 |
| **Dev (10%)** | 3,107 | 1,513 | 4,567 | 275 | 2,475 | 323 |
| **Test (10%)** | 2,945 | 1,587 | 5,012 | 360 | 2,514 | 271 |

Table 4: The number of grouped comments in AOC-ALDi's splits. 127,835 comments of 20 words on average, are distributed across all splits.

| Comment | English translation (ours) | ALDi |
|---|---|---|
| برافو للسيد الوزير الرائع الذي اثبت انه مسئول يشعر بمدى أهميه مسئوليته لاول مره منذ زمن بعيد في تاريخ التعليم المصري | Bravo to the wonderful Minister, who proved that he is responsible, feeling the importance of his responsibility for the first time in a long time in the history of Egyptian education. | $0, 0, \frac{1}{3}$ $\approx 0.11$ |
| نبتدى بقى الشغل الصح فى تطوير المدارس وتوفير المراقبين عليها | We start with the right task of developing schools and providing observers over them | $\frac{1}{3}, \frac{1}{3}, 1$ $\approx 0.56$ |
| وزير جدع بصراحة ... ياريت يفضل كدا على طول | Honestly, a serious minister .... I hope he stays like this all the time | $\overline{1, 1, 1}$ $\approx 1.00$ |

Table 5: Sample comments to the same article with their level of dialectness labels (3 annotations for each comment with their $\overline{mean}$ as the ALDi score). The labels are *MSA* (0), *Little* ($\frac{1}{3}$), *Mixed* ($\frac{2}{3}$), *Most* (1). DA segments are underlined. Loanwords are double-underlined.

ues. Although these agreement levels are considered only moderate, our experiments demonstrate that the corpus can nevertheless be useful.

**Step #3 - Label aggregation**: Multiple human annotations for the level of dialectness were aggregated into a single label. We transformed the ordinal labels (MSA, Little, Mixed, Mostly) into the numeric values (0, $\frac{1}{3}$, $\frac{2}{3}$, 1), then took the algebraic mean of these as the gold standard label, which has the range $[0, 1]$.[5] The distribution of the aggregated scores across four intervals is shown in Figure 2.

**Step #4 - Splits creation**: To build reliable splits of AOC, we made sure comments to the same document are in the same split. For each source, we group sentences belonging to the same article together, shuffle these groups, and then assign the first 80% of the comments to the training split, the following 10% to the development split, and the remaining 10% to the test split. This way, the dev and test sets evaluate whether a model generalizes to comments from articles not seen in training. The distribution of the sources across AOC-ALDi's splits is in Table 4.

**Qualitative Analysis**: Table 5 shows three example sentences from the AOC-ALDi dataset with their corresponding annotations where all annotators labeled the dialect as either MSA, EGY, or GEN. The first sentence begins with an English

loanword. The rest of the sentence has MSA terms that will not sound natural if pronounced according to the phonetic rules of a variant of DA. Unsurprisingly, two annotators considered the sentence to be in MSA, while the third might have perceived the presence of the loanword as a sign of dialectness, thus marking the sentence as *little* dialectal. The second example shows code-switching between MSA and Egyptian DA, but an Egyptian can still naturally pronounce the MSA portion abiding by the phonetic rules of Egyptian Arabic. This might be the reason why one of the annotators labeled the sentence as mostly dialectal (see Parkinson (1991), who observed the same relation between pronunciation and perceived levels of dialectness). For the third example, all the tokens except for the first one show dialectal features, which made it easy for the three annotators to classify it as *mostly* dialectal.

## 4 The ALDi Estimation Task

Before describing case studies demonstrating possible uses of automatic ALDi estimation, we first show that a model trained to predict ALDi is competitive with a DI system in discriminating between dialects (including dialects barely represented in AOC-ALDi), while providing more nuanced dialectness scores. We then consider several specific features of Egyptian Arabic, and again show that the ALDi regression model is more sensitive to these than the baseline approaches.

---

[5]AOC-ALDi also includes the original separate labels.

## 4.1 Models

The main model we use to predict ALDi is a BERT-based regression model. Using the training split of *AOC-ALDi*, we fine-tune a regression head on top of MarBERT, an Arabic BERT model (Abdul-Mageed et al., 2021a), and clip the output to the range $[0, 1]$. To measure the consistency of the model's performance, we repeat the fine-tuning process three times using 30, 42, and 50 as the random seeds, and report averaged evaluation scores for the model (similarly for Baseline #3). We compare this model to three baselines, which use existing Arabic resources and are not trained on AOC-ALDi.

**Baseline #1 - Proportion of tokens not found in an MSA lexicon**: The presence of dialectal lexical terms is one of the main signals that humans use to determine dialectal text. Sajjad et al. (2020) built an MSA lexicon from multiple MSA corpora. They then computed the percentage of tokens within a sentence not found in the MSA lexicon as a proxy for sentence-level dialectness. We replicate this method using the tokens occurring more than once in the Arabic version of the United Nations Proceedings corpus (Ziemski et al., 2016) as the source for the MSA lexicon.

**Baseline #2 - Sentence-Level DI**: We use an off-the-shelf DI model implemented in (Obeid et al., 2020) based on (Salameh et al., 2018). The model is based on Naive Bayes, trained on the MADAR corpus (Bouamor et al., 2018), and uses character and word $n$-grams to classify a sentence into 6 variants of DA in addition to MSA. A sentence is assigned an ALDi score of 0 if it is classified as MSA and a score of 1 otherwise.

**Baseline #3 - Token-level DI**: Molina et al. (2016) created a token-level DI dataset *(MSA-EGY token DI)*, in which tokens of tweets were manually tagged as MSA, EGY, Named-Entity, ambiguous, mixed, or other. We use this dataset to fine-tune a layer on top of MarBERT to tag tokens of a sentence. The tag of the first subword for each token is adapted as the tag for the whole token as done in (Devlin et al., 2019). We use token-level tags to compute the Code-Mixing Index (CMI; Das and Gambäck 2014) as a proxy for ALDi: $CMI = \frac{N_{EGY\ tokens}}{N_{EGY\ tokens} + N_{MSA\ tokens}}$ (set to 0 if none of the tokens are tagged as MSA or EGY).

## 4.2 Evaluation

**Intrinsic AOC-ALDi evaluation**    Treating the aggregated human-assigned scores of AOC-ALDi's

| Model | $RMSE(\downarrow)$ | | |
| --- | --- | --- | --- |
| | **Cntrl** N=2,127 | **Cmnt** N=10,644 | **All** N=12,771 |
| **MSA Lexicon** | 0.13 | 0.36 | 0.34 |
| **Sentence DI** | 0.23 | 0.53 | 0.49 |
| **Token DI** | 0.11* | 0.33* | 0.30* |
| **Sentence ALDi** | **0.07*** | **0.19*** | **0.18*** |

Table 6: Models' RMSE on AOC-ALDi's test split. *: Average values across three fine-tuned models with different random seeds. The corresponding standard deviations are 0.015 or less.

test split as the gold standard, we measure how the models' ALDi predictions deviate from the gold standard ones using Root Mean-Square Error (RMSE). As expected since it is the only model trained on AOC-ALDi, the *Sentence ALDi* model achieves the least RMSE of 0.18 on the AOC-ALDi test split, as indicated in Table 6. The two other models that can produce continuous scores at the sentence level, *MSA Lexicon* and *Token DI*, achieve similar RMSE, and are both better than the binary *Sentence DI* model despite more limited exposure to the dialects in this corpus (recall that *Token DI* has only been trained on EGY and MSA, and *MSA Lexicon* has no explicit DA training). All models perform worse on the comments than the controls.

**Disentangling Parallel MSA/DA Sentences**    For a model estimating ALDi, a minimal requirement is to assign a higher score to a DA sentence than that assigned to its corresponding MSA translation.

We utilize two parallel corpora of different genres and dialects to test this requirement. First, we use a parallel corpus of 8219 verses (sentences) from the **Bible**, provided by Sajjad et al. (2020), which includes versions in MSA, Tunisian, and Moroccan Arabic. We also use **DIAL2MSA**, which is a dataset of dialectal Arabic tweets with parallel MSA translations (Mubarak, 2018). Five MSA translations were crowd-sourced for 12,000 tweets having distinctive lexical features of Egyptian and Maghrebi Arabic. Each translation was then validated by 3 judges. For our analysis, we discard samples having a non-perfect validation confidence score, and ones that still have a distinctive dialectal lexical term in their MSA translations.

The distribution of the ALDi scores in Figure 3 reveals that *MSA Lexicon* does not discriminate strongly between MSA and DA, while *Token DI* mostly assigns scores of 0 or 1 (acting like *Sentence DI*), despite the possibility to do otherwise.

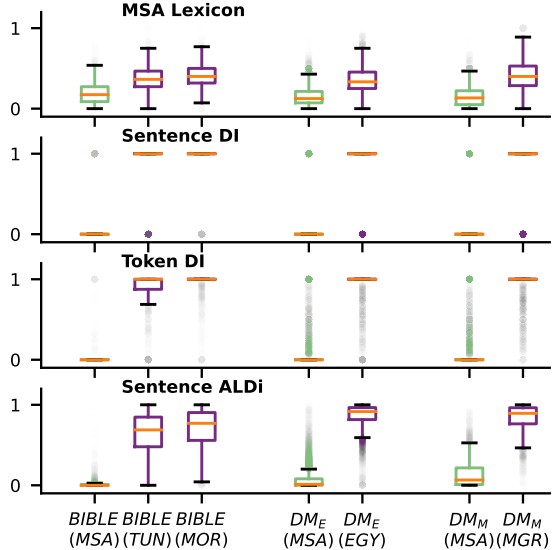

Figure 3: The distribution of the ALDi scores assigned by the four models to sentences of the Bible and DIAL2MSA corpora. Each column (across the four plots) represents the same set of sentences as scored by the four different models, and the columns are grouped by corpus to compare the different dialectal versions of that corpus. For each plot, the orange line shows the median score, the box represents the interquartile range (IQR) $[Q1, Q3]$ of the scores, the whiskers represent $\pm 1.5 * \Delta(IQR)$ beyond Q1 and Q3, and the dots represent outliers beyond this. **Note1**: $\Delta(IQR) = Q3 - Q1$. **Note2**: The boxplots for the *Token DI* and *Sentence ALDi* models are not significantly different across the multiple fine-tuning runs of different random seeds.

The *Sentence ALDi* model provides more nuanced scores while also showing strong discrimination between MSA and DA, even for DA variants that are barely present in AOC-ALDi (TUN, MOR, MGR; note that *Token DI* also has not seen these).[6] It also yields slightly lower scores for the DA versions of the Bible than for the DA tweets, indicating that the informal genre of tweets may be an indicator of stronger dialectness levels.

### 4.3 Analysis - Minimal Contrastive Pairs

Inspired by Demszky et al. (2021)'s corpus of minimal contrastive pairs for 18 distinctive features of Indian English, we build contrastive pairs of MSA and Egyptian Arabic variants of a single sentence. We investigate 5 features of Egyptian Arabic that were previously recognized by Darwish et al. (2014). For each sentence, we generate versions with different gender markings (masculine

---

[6]Further discussion, including $D'$ discrimination scores for all models, can be found in Appendix C.

and feminine) and word orders (SVO and VSO). While MSA allows for both word orders, it favors VSO (El-Yasin, 1985), while Egyptian Arabic favors SVO (Gamal-Eldin, 1968 as cited in Holes, 2013; Zaidan and Callison-Burch, 2014). In Table 7, we display the ALDi scores assigned by the different models to the contrastive pairs.

The *MSA Lexicon* model considers all dialectal features to have the same impact in assigning a non-zero ALDi score (i.e., $\frac{1}{3} \approx 0.33$ or $\frac{1}{2} \approx 0.5$) to the DA sentences. As implied by our previous experiment, the *Token DI* model acts as a sentence-level DI model, tagging all the tokens as dialectal if only one token shows a distinctive dialectal feature. This behavior might be an artifact of the model's fine-tuning dataset, where annotators were asked to use the surrounding context to determine an ambiguous token's language (EGY or MSA).

Conversely, the *Sentence ALDi* model provides a more nuanced distinction between the different features. The negation form (F4, F5) used in Egyptian Arabic seems to cause the model to categorically consider the sentence as highly dialectal. Less salient features such as the (F1) present progressive proclitic بـ increase the ALDi level of the sentence, but to a lesser extent than the negation feature. We also see that the model assigns higher ALDi scores to SVO sentences than VSO, suggesting that the model may have learned the common word order in Egyptian Arabic. Finally, feminine-marked sentences tend to get higher scores compared to their masculine-marked counterparts, which may be indicative of a gender bias in the training data and resulting model—if feminine marking is less common, it may also be seen as less standard language and interpreted as non-MSA.

## 5 Case Studies (ALDi in Practice)

The same speaker can adapt different styles according to various social and linguistic factors (Kiesling, 2011). The ALDi of speech is one example of an intraspeaker variation in Arabic. In this section, we provide two case studies analyzing the transcribed speeches of three different Arab presidents. We highlight how quantitatively estimating the ALDi can help in revealing different speaking styles.

### 5.1 Presidential Speeches in the Arab Spring

Lahlali (2011) qualitatively analyzed the usage of MSA and DA (Tunisian Arabic and Egyptian Arabic) in the last three speeches of Ben-Ali and

| Feature name | MSA$_f$ | EGY$_f$ | Word order | MSA LEX | | SEN. DI | | TOK. DI * | | SEN. ALDi * | |
|---|---|---|---|---|---|---|---|---|---|---|---|
| | | | | MSA | EGY | MSA | EGY | MSA | EGY | MSA | EGY |
| **F1) Present progressive** | تقول البنت الحقيقة | بتقول البنت الحقيقة | VSO | 0.0 | 0.33 | 0.0 | 0.0 / 1.0 | 1.0 | 1.0 | 0.1 / 0.12 | 0.86 / 0.56 |
| **En**: The girl is saying the truth | البنت تقول الحقيقة | البنت بتقول الحقيقة | SVO | 0.0 | 0.33 | 0.0 | 1.0 | 1.0 | 1.0 | 0.23 / 0.26 | 0.83 / 0.62 |
| **F2) Future Morpheme** | ستقول البنت الحقيقة | هتقول البنت الحقيقة | VSO | 0.0 | 0.33 | 0.0 | 0.0 | 0.0 | 1.0 | 0.0 / 0.07 | 0.76 / 0.9 |
| **En**: The girl will say the truth | البنت ستقول الحقيقة | البنت هتقول الحقيقة | SVO | 0.0 | 0.33 | 0.0 | 1.0 / 0.0 | 0.11 / 0.0 | 1.0 | 0.02 / 0.09 | 0.79 / 0.89 |
| **F3) Passive formation** | قيلت الحقيقة | اتقالت الحقيقة | VO | 0.0 | 0.5 | 0.0 | 0.0 | 0.0 | 1.0 | 0.05 | 0.36 |
| **En**: The truth was said | الحقيقة قيلت | الحقيقة اتقالت | OV | 0.0 | 0.5 | 0.0 | 1.0 | 0.0 | 1.0 | 0.11 | 0.36 |
| **F4) Negation** | لا تقول البنت الحقيقة | مبتقولش البنت الحقيقة | VSO | 0.0 | 0.33 | 0.0 | 1.0 | 0.17 / 0.0 | 1.0 | 0.08 / 0.11 | 0.95 / 0.91 |
| **En**: The girl is not saying the truth | البنت لا تقول الحقيقة | البنت مبتقولش الحقيقة | SVO | 0.0 | 0.33 | 0.0 | 1.0 | 0.25 / 0.33 | 1.0 | 0.09 / 0.11 | 0.91 / 0.9 |
| **F5) Negated imperative** **En**: Do not say the truth | لا تقولي الحقيقة | ماتقوليش الحقيقة | VSO | 0.0 / 0.33 | 0.5 | 0.0 | 1.0 | 0.0 | 1.0 | 0.0 / 0.13 | 0.84 / 0.91 |

Table 7: The ALDi scores assigned to contrastive MSA and Egyptian Arabic sentences. Only the feminine-marked version of the sentence is shown, and tokens with dialectal features are underlined. A single score is reported if a model assigns the same score to the masculine and feminine versions of a sentence, otherwise the scores for masculine/feminine are shown. We tested VSO (favored in MSA) and SVO (favored in EGY) word orders.
**Note:** Scores $\in [0, 0.11]$ are encoded in green, while ones $\in\, ]0.11, 1]$ have a shade of purple.
*: The scores for these models are averaged across three fine-tuned models with different random seeds.

Mubarak, the former Tunisian and Egyptian presidents during the period of the Tunisian and Egyptian revolutions. Mubarak consistently used MSA for his speeches to showcase authority and power. Ben-Ali used MSA for his first two speeches. For his last speech, he explicitly said: "نكلمكم لغة كل التونسيين والتونسيات" - "I talk to you in the language of all the Tunisians", apparently using his choice of dialect as a way to identify himself with a particular group (cf. Shoemark et al. 2017; McNeil 2022).

We quantitatively replicate the analysis by visualizing the ALDi scores of the transcribed speeches. We scraped the speeches from online websites[7] and used the HTML line breaks *
* to segment them into sentences. For each sentence, we predict the ALDi score with our model and also use the *Sentence DI* model to classify it as DA or MSA.

Figure 4a shows that our model correctly finds nearly 0 ALDi scores throughout Ben-Ali's speech on the 10[th] of January, while the DI model makes a couple of errors (and similarly for Mubarak's speeches, shown in Figures 4c, 4d). Both models identify the shift to DA in the second speech (Figure 4b), with more sentences identified as DA by the DI model, and many with moderate ALDi scores. Given the nature of the speech, Ben-Ali still used formal terms while speaking in Tunisian Arabic which is likely the reason for the intermediate ALDi scores.

### 5.2 El-Sisi's Speeches

Next, we studied the ALDi scores for 659 speeches of the current Egyptian president El-Sisi, scraped

from `almanassa.com`. The transcripts are not limited to the edited presidential speech, but also include greetings, introductory comments, interventions by the audience, and signs of disfluency or hesitation. The site's editors segmented each speech into coherent sentences, embedded in *<p>* HTML tags, that we adapt as units of analysis.

While most of these speeches are conducted in MSA, multiple cases of code-switching between MSA and Egyptian Arabic occur. For example, in Figures 4e and 4f, El-Sisi used MSA when reading the edited speech, and Egyptian Arabic with high ALDi scores when spontaneously addressing the audience before or after the edited speech.

Interestingly, Figure 4g shows three different ALDi levels as part of the same speech. El-Sisi used MSA for reading the edited speech directed to the press, discussing issues such as Egyptian-German diplomatic relations, climate change, and economic hardships. He then reacted spontaneously to two questions from the press. He attempted to answer the first question, related to gas prices, in MSA but the sentences show code-switching between MSA and Egyptian Arabic, indicated by intermediate ALDi scores (though the DI system does not identify these). For the second question about human rights in Egypt, El-Sisi uses sentences that are more dialectal and less formal, inviting the journalist to visit Egypt in order to make a fair assessment of the situation. This is indicated by even higher ALDi scores. Samples from each segment are listed in Appendix D.

This speech is a clear example of how an Arabic speaker can adapt different levels of dialectness in their speech and indicates the ability of ALDi to reveal such differences.

---

[7] `www.babnet.net` and `egypt-blew.blogspot.com`

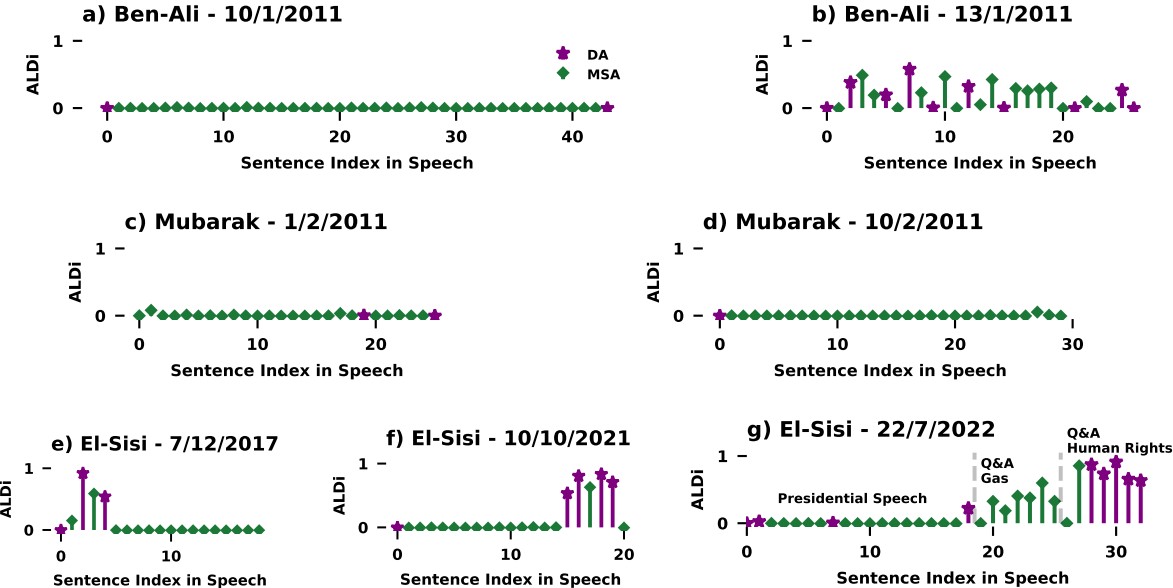

Figure 4: The ALDi scores assigned to sentences of transcribed political speeches. Subfigures a) and b) represent two speeches of the former Tunisian president Ben-Ali during the Tunisian Revolution. Subfigures c) and d) represent two speeches of the former Egyptian president Mubarak during the Egyptian Revolution. Subfigures e), f), and g) are speeches of the current Egyptian president El-Sisi. The MSA/DA labels were generated by the *Sentence DI* model.

## 6 Conclusion

We presented ALDi, a linguistic variable that quantifies the level of dialectness of an Arabic sentence. We release AOC-ALDi, a dataset of Arabic comments annotated with their ALDi scores. A BERT-based regression model fine-tuned on AOC-ALDi showed superior performance compared to existing baselines that are based on lexicons and DI models. Our analysis shows that the model generalizes to various Arabic dialects. In addition, the model provides a nuanced distinction of dialectal features, which token and sentence DI models can not perform. Lastly, we presented multiple case studies demonstrating the effectiveness of ALDi in revealing new insights in Arabic text. For future work, we aim to explore the possible applications of ALDi for text analysis, especially for sociolinguistics and computational social science studies. Moreover, we aim to apply the level of dialectness work to other languages that have the same phenomena of Arabic, such as Swiss-German.

## Limitations

Our AOC-ALDi dataset is based on the AOC dataset that comes mainly from news comments, which might be of specific genre. Although our ex-

periments show robustness across multiple genres of text, it will be interesting to prepare a dataset (even just for intrinsic testing) that comes from other sources, such as social media. Reannotating existing DI datasets with ALDi might be a first-to-do option.

Also, the gold-standard ALDi scores in our AOC-ALDi dataset are based on normalizing the level of dialectness annotations of the AOC dataset, which might be sub-optimal. Labeling a dataset directly with continuous ALDi scores might provide more accurate labels (still might be more challenging for annotators).

While our experiments cover diverse dialects of Arabic, the generalizability of ALDi for more dialects of Arabic more dialects needs to be tested.

Finally, we found preliminary evidence of possible gender bias in our dataset/model. While we did not explore this issue in depth here, it will be important to consider its impact and possible mitigation strategies in future work.

## Acknowledgments

This work was supported by the UKRI Centre for Doctoral Training in Natural Language Processing, funded by the UKRI (grant EP/S022481/1) and the

University of Edinburgh, School of Informatics.

We thank Nizar Habash for his valuable input regarding his previous efforts related to the idea of the level of dialectness.

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

## A  AOC Annotation Interface

Zaidan and Callison-Burch (2011) used Amazon Mechanical Turk to annotate Arabic comments they scraped from three different newspapers. They provided the annotators with minimal guidelines for determining the dialect and level of dialectness of the comments. A screenshot of their annotation interface is shown in Figure A1.[8]

While the guidelines are minimal, we think that the Arabic and English translations of the labels might have impacted the annotator's understanding of the labeling process. For instance, the annotation interface has the *Not Arabic* label translated to (لغة أخرى أو رموز) in Arabic, which actually means

---

[8]The annotation site can be accessed through `https://www.cs.jhu.edu/data-archive/RCLMT-2011/html/dialect_classification.shtml`.

# Help Classify Arabic into Dialects!

This task is for Arabic speakers who understand the different local Arabic dialects (اللهجات العامّية، أو الدّارجة), and can distinguish them from *Fusha* Arabic (الفصحى).

Below, you will see several Arabic sentences. For each sentence:

1. Tell us how much dialect (عامّية) is in the sentence, and then
2. Tell us which Arabic dialect the writer intends.

This following map explains the dialects:

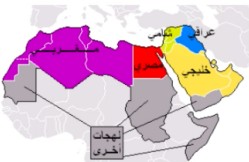

PLEASE READ the following. You MUST understand the classifications, otherwise your work might be rejected!!

- **Levantine** (شامي) does NOT mean "Syrian" only. It includes **Syrian**, but ALSO: **Jordanian** is Levantine, **Palestinian** is Levantine, and **Lebanese** is Levantine. That's why all these countries are **green** in the map.

- **Maghrebi** (مغربي) does NOT mean "Moroccan" only. It includes **Moroccan**, but ALSO: **Algerian** is Maghrebi, **Tunisian** is Maghrebi, and **Libyan** is Maghrebi. That's why all these countries are **purple** in the map.

- The word "dialect" (لهجة) does NOT mean "spelling mistake" (خطأ إملائي). If the writer was trying to write in 100% فصحى, classify it as **No dialect**, even if it has some spelling mistakes.

**This is a simple task, and your answers will help advance research on the Arabic language, so please do the task properly, and please have fun doing it. :)**

First, please answer these questions about your language abilities:
***You don't have to answer these questions in every HIT; one time is enough)***

Is Arabic your native language? ○ Yes ○ No
How many years have you spoken Arabic? (If native speaker, just enter your age.) [____] years
Which Arabic dialect do you understand best? [Choose dialect... ⌄]
What country do you currently live in? [________]

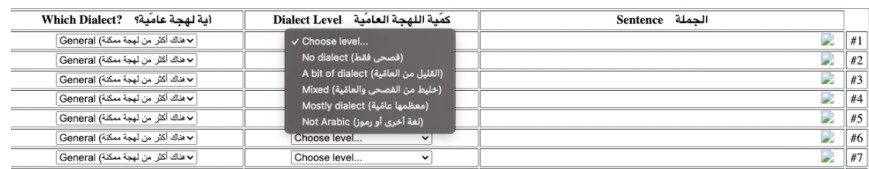

Figure A1: A screenshot of the annotation interface of the AOC dataset (Zaidan and Callison-Burch, 2011).

| Reason for Discarding | Sentence | Source | Level of Dialectness |
|---|---|---|---|
| **Symbols** | ؟؟؟؟؟ | Cmnt (Y7) | ¬ Arabic (x13), Missing (x2) |
| | ******** | Cmnt (Ri) | ¬ Arabic (x3) |
| **English** | gloves to protect the baby from infection !ممكن تلبس | Cmnt (Ri) | ¬ Arabic (x2), MSA (x1) |
| | I agree with you that racism exists in the United States; I also know it exists in Arab countries as well. Just remember that America elected a black president with 360 electoral college votes. In terms of numbers, that means a sweeping majority. Lets learn to be better than the Americans by developing our own democratic systems for a change...ccc very nice... | Cmnt (Gh) | ¬ Arabic (x3) |
| | | Cmnt (Ri) | ¬ Arabic (x3) |
| **Arabizi** | ya zamalek ya 7arameyaaaa | Cmnt (Y7) | ¬ Arabic (x2), Most (x1) |
| | ma howeh el blogs m3abbiyeh el denya ? ya3ni law doctor el jam3a bedo yet3ab shway w yekteb articles, ma kan 3emel blog men zaman. | Cmnt (Gh) | ¬ Arabic (x2), Most (x1) |
| **URLs and Emails** | http://elbeet-elmuslim.ace.st/forum.htm | Cmnt (Y7) | ¬ Arabic (x3) |
| | Ahmad.altamimi@alghad.jo | Cntrl (Gh) | ¬ Arabic (x3) |
| **Presence of HTML** | █████ 5000 √DONE | Cmnt (Y7) | ¬ Arabic (x3) |
| | د. أشرف بلبع<"a href="EditorOpinions.asp?EditorID=404> </a> | Cntrl (Y7) | ¬ Arabic (x3) |
| | ١٩٨١ بيتهيألى قربنا قوى من سبتمبر | Cmnt (Y7) | ¬ Arabic (x2), Most (x1) |

Table A1: Examples of the discarded AOC comments of majority labels set to Not Arabic or missing.
**Note**: **Cmnt** stands for comment, **Cntrl** stands for control sentence, **Y7**: Youm7, **Ri**: AlRiyadh, **Gh**: AlGhad.

(*Another language, or symbols*). We believe that *Another language or symbols* is not equivalent to *Not Arabic*, which might make annotators interpret the guidelines differently.

## B  Discarded Samples from AOC-ALDi

As mentioned in §3.2, we discarded 2,038 comments that have the majority of their ALDi annotations either set to *Not Arabic* or are missing. Five different categories of such comments were identified as per Table A1. These categories include sentences that have only punctuation marks, are written in English or Arabizi (Romanized Arabic (Yaghan, 2008)), are just links to sites or emails, or have HTML encoded characters or formatting tags.

## C  Discrimination scores

For the experiments in §4.2, we computed $D'$, a measure of discrimination, for all models on each pair of parallel corpora. Results are shown in Table C2. On the DIAL2MSA corpora, which are likely more similar in style to AOC-ALDi, our model performs about as well as *Token DI*, the other BERT-based model (which, like ours, has not seen MGR in training), while also providing a wider range of scores (as shown in §4.2). *Token DI* does somewhat better than our model on the Bible corpora, but again by making nearly binary judgments for each sentence.

## D  Edited and Spontaneous Speech

As depicted in Figure 4g, El-Sisi's speech on the $22^{nd}$ of July 2022 can be split into three segments: the edited presidential speech, and two spontaneous responses from the president to questions from the succeeding press conference.

We sampled a sentence from each segment as shown in Table D3 to demonstrate the three different levels of dialectness that categorize each segment.

| Model | Bible | | DIAL2MSA | |
|---|---|---|---|---|
| | MSA / TUN | MSA / MOR | MSA / EGY | MSA / MGR |
| **MSA Lexicon** | 1.28 | 1.55 | 1.48 | 1.73 |
| **Sentence DI** | 2.65 | 3.89 | 2.17 | 2.76 |
| **Token DI\*** | **3.81 ± 0.26** | **5.56 ± 0.34** | **5.83 ± 0.13** | 3.93 ± 0.03 |
| **Sentence ALDi\*** | 3.35 ± 0.09 | 3.89 ± 0.25 | 5.16 ± 0.13 | **4.15 ± 0.1** |

Table C2: The $D'(\uparrow)$ scores for the parallel MSA/DA corpora. **TUN**: Tunisian Arabic, **MOR**: Moroccan Arabic, **EGY**: Egyptian Arabic, **MGR**: Maghrebi Arabic.
\*: $D'$ scores averaged across three fine-tuned models with different random seeds (30, 42, 50).

| Segment | Sentence with English translation (ours) | ALDi (estimated) |
|---|---|---|
| Main speech | واتفقنا على أن الوضع الحالي يفرض على كافة الفاعلين الدوليين التحلي بالمسؤولية لإيجاد حلول وآليات عملية تخفف من تداعيات الأزمة على الدول الأكثر تضررًا. | 0 |
| | And we agreed the current circumstances endeavors all actors to bear their responsiblities by finding practical solutions and mechanisms to mitigate the impact of the crisis on the most affected countries. | |
| Q&A - Gas | وأنا عايز أقول إن إحنا تحدث، يعني أنا تحدثت في هذا الأمر إن المطلوب التنسيق والتعاون بين كل دول العالم في ما يخص هذا الملف أثناء حديثي أو خطابي في مؤتمر جدة عن موضوع الطاقة تحديدًا. | 0.41 |
| | I spoke about this matter and that coordination and cooperation are required between all countries of the world regarding this topic during my talk or speech at the Jeddah conference, specifically on the issue of energy. | |
| Q&A - Q&A - Human Rights | وإحنا مش مهتمين بيه عشان أنتوا بتسألوا عنه.. مهم قوي إن إنتوا تعرفوا كدا. إحنا مهتمين بيه عشان إحنا بنحترم شعوبنا، وبنحبها، ومش كلام، إحنا بنحترم شعوبنا زي ما إنتوا ما بتحترموا شعوبكم.. وبالتالي إحنا مش مهتمين عشان أنتوا بتسألونا عليه.. لأ.. ده مسؤوليتنا الأخلاقية والتاريخية والإنسانية تجاه شعوبنا. دي نقطة. | 0.75 |
| | And we are not interested in it because you ask about it.. It is very important that you know this.. We are interested in it because we respect our people, and we love them, and these are not just words, we respect our people just as you respect your people.. and therefore we are not interested because you ask us about it. .. No.. This is our moral, historical and humanitarian responsibility toward our people. This is one point. | |

Table D3: Three sentences of different estimated ALDi scores sampled from three segments of El-Sisi's speech on the 22$^{nd}$ of July 2022 shown in Figure 4g.