# OpenReview forum: "ALDi: Quantifying the Arabic Level of Dialectness of Text"
_EMNLP/2023/Conference — EMNLP 2023 Main_

### Official Review · Reviewer_t5z9 · 2023-08-04

**Typos Grammar Style And Presentation Improvements:** N/A
**Soundness:** 4

**Excitement:**

4: Strong: This paper deepens the understanding of some phenomenon or lowers the barriers to an existing research direction.

**Missing References:**

N/A

**Paper Topic And Main Contributions:**

The paper presents a new contribution to the area of Arabic Dialect Identification. Specifically the authors manually annotate a large corpus for a different degrees of dialectness at the sentence level and define what they name as the ALDi Score.   The intuition that dialectness in sentences can be represented on a continuum has been expressed in the past, but this work does a nice job of quantifying and studying and producing data sets to support this intuition.

**Questions For The Authors:**

1. in Table 5 - ALDi scores are of three annotators? it's not clear from the caption.

2. Was every sentence labelled thrice?

3. Is the ALDi predicted label always in the set of {0, 1/3, 2/3, 1} or is a value between 0 and 1?
And when you evaluate, do you compute the RMSE against the average or against the closest value of the multiple annotators?

4. Figure A1 is not showing the sentences (broken image links appear)

5. Have you considered using morphological analysis and disambiguation tools from Camel Lab... they exist for MSA and EGY... in principle you can use the ration of OOV against different analyzers.

6. I assume you do not use MADAR in your annotations because of Baseline #2 which uses this data. perhaps clarify?

7. Camel Tools dialect ID produces a vector of scores for MSA and dialects... why not compare against them?

**Reasons To Accept:**

* The work presents a novel contribution to the field of Dialect Identification in general and Arabic specifically.
* The effort is extensive
* The paper is well written
* The data is public

**Reasons To Reject:**

* The agreement among annotators is only moderate.
* One of the baselines used (#2, using Salameh/CamelTools) only uses the binary reading of the output; but the tool actually produces a vector of scores... why not compare against them?  Try the interface here: https://adida.abudhabi.nyu.edu/ to see the subscores.

**Reproducibility:**

5: Could easily reproduce the results.

**Reviewer Confidence:**

5: Positive that my evaluation is correct. I read the paper very carefully and I am very familiar with related work.

---

> ### Author Rebuttal · Authors · 2023-08-24
>
> We are grateful to Reviewer t5z9 for thoroughly checking our paper and for their valuable questions and suggestions. Kindly find our responses below:
>
> * **Q1**: Yes, each sentence has three annotations. We will update the caption of Table 5 to make it clear.
>
> * **Q2**: Yes, every sentence was labelled thrice.
>
> * **Q3**: The fine-tuned model generates a continuous ALDi score in the range [0, 1]. For the evaluation, the RMSE is computed between the averaged human-assigned scores and the predicted value. We will make sure this is clearly mentioned in the captions of tables of figures (e.g.: Table 5, Figure 3) to avoid any confusion.
>
> * **Q4**: This screenshot is for the interface used by (Zaidan and Callison-Burch, 2011) [1], so we are afraid the images are not recoverable.
>
> * **Q5**: Thanks for the suggestion. While it would be one more good baseline to test, we think that the ability of this baseline to generalize to other lower-resourced dialects of Arabic would require building morphological analyzers for these dialects.
>
> * **Q6**: As mentioned in lines 185-196, we analyzed the “level of dialectness” labels that are already part of the Arabic Online Commentary (AOC) dataset which precedes the MADAR corpus. We will be sure that this is mentioned clearly in the final version of the paper.
>
>
> * Regarding the suggestion to **use the vector of scores from a Dialect Identification (DI) model** (+**Q7**), we tested the idea using the following equation **Estimated ALDi score(text) = 1 - MSAConfidenceScore(text)**, where MSAConfidenceScore is the confidence score that the Sentence DI assigns to the MSA label. The RMSE values shown in the table below show that using the confidence scores can slightly improve the baseline (RMSE 0.46 vs 0.49), however, it is still significantly worse than our proposed model (0.19).
>
> | Model | RMSE(Control Sentences) (N=2,127) | RMSE(Comments) (N=10,644) | RMSE(All) (N=12,771) |
> | ---: | :---: | :---: | :---: |
> | (old baseline) Sentence DI | 0.27 | 0.53 | 0.49 |
> | (suggested baseline) Sentence DI + Confidence scores | 0.27 | 0.48 | 0.46 |
> | (our proposed method) Sentence ALDi | 0.07 | 0.2 | 0.19 |
>
>   * Our intuition is that mapping the confidence scores that a Sentence DI model assigns to specific dialects vs. MSA is inadequate for estimating the ALDi score of a sentence. These confidence scores reflect the ability of the DI model to determine the specific dialect of a sentence. A model can be really confident about the dialect of the sentence, even if it only shows a single cue of dialectness that is quite distinctive for one dialect. However, a single cue of dialectness would not generally make the whole sentence highly dialectal (i.e.: the ALDi score should not be high).
> Again, we thank you for the suggestion, and we will update the baseline in the paper to the one you suggested along with this discussion.
>
>
>
>
>
>
> ***
> [1] Omar F. Zaidan and Chris Callison-Burch. 2011. The Arabic Online Commentary Dataset: an Annotated Dataset of Informal Arabic with High Dialectal Content. In Proceedings of the 49th Annual Meeting of the Association for Computational Linguistics: Human Language Technologies, pages 37–41, Portland, Oregon, USA. Association for Computational Linguistics. (https://aclanthology.org/P11-2007/)

---

### Official Review · Reviewer_RifN · 2023-08-05

**Paper Topic And Main Contributions:** 1- The paper proposes a new approach …
**Soundness:** 4

**Excitement:**

4: Strong: This paper deepens the understanding of some phenomenon or lowers the barriers to an existing research direction.

**Reasons To Accept:**

1- The authors introduce the concept of Arabic Level of Dialectness (ALDi), a continuous linguistic measure to capture the nuanced spectrum of dialect usage, shifting the focus from binary dialect identification to a continuous spectrum of dialectness.

2-They establish the AOC-ALDi dataset, offering a robust foundation for evaluating this concept. The paper highlights that ALDi offers a more nuanced perspective compared to binary dialect identification systems.

3- The paper's exploration of ALDi can inspire researchers to apply similar approaches to other languages.

**Reasons To Reject:**

The paper could provide more details on how the proposed approach can be replicated, potentially offering guidelines for researchers interested in applying the ALDi framework to other languages

**Reproducibility:**

4: Could mostly reproduce the results, but there may be some variation because of sample variance or minor variations in their interpretation of the protocol or method.

**Reviewer Confidence:**

4: Quite sure. I tried to check the important points carefully. It's unlikely, though conceivable, that I missed something that should affect my ratings.

---

> ### Author Rebuttal · Authors · 2023-08-24
>
> We really thank Reviewer RifN for their interest in the paper and their positive feedback.

---

### Official Review · Reviewer_sVaX · 2023-08-05

**Soundness:** 3

**Excitement:**

3: Ambivalent: It has merits (e.g., it reports state-of-the-art results, the idea is nice), but there are key weaknesses (e.g., it describes incremental work), and it can significantly benefit from another round of revision. However, I won't object to accepting it if my co-reviewers champion it.

**Paper Topic And Main Contributions:**

The paper tackles the problem of Dialect Identification in Arabic by annotating an existing corpus to add "level of dialectalness" labels to it. The existing dataset comprised comments crawled from several Arabic websites and annotated each comment as either being Dialectal or Standard Arabic along with the dialect (e.e. EGY, LEV, other) it belongs to. The authors fine-tune a BERT-based model in the dataset and compare the performance to different baselines (e.g., ones that measure the number of tokens present in or absent from dictionaries) and find that the fine-tuned model outperforms the baselines. While the paper tackles an important problem for Arabic, the proposed dataset only presents a marginal contribution over existing Dialectal Arabic resources.

**Questions For The Authors:**

- In Table 1, how do you distinguish between the level of dialectalness and the level of formality? The distinction between the examples for the medium and high level of dialectalness is that the later is more informal (not necessarily more dialectal). I think higher level of dialectalness should be more related to how much of the given input is dialectal versus standard.

**Reasons To Accept:**

The studied problem (code switching in Arabic between the standard form and the dialectal variants) is an interesting one for Arabic language

**Reasons To Reject:**

The proposed dataset only presents a marginal contribution over existing Dialectal Arabic resources and it is not clear that the task itself (i.e. level of dialectalness) is well defined since it conflates formality with dialectalness.

**Reproducibility:**

3: Could reproduce the results with some difficulty. The settings of parameters are underspecified or subjectively determined; the training/evaluation data are not widely available.

**Reviewer Confidence:**

5: Positive that my evaluation is correct. I read the paper very carefully and I am very familiar with related work.

---

> ### Author Rebuttal · Authors · 2023-08-24
>
> We thank the Reviewer sVaX for raising the question about the relationship between the “formality” and the “level of dialectness” concepts. We have thoroughly thought about this relationship while working on the paper, and dedicated a section to discussing why they are different, and why the "level of dialectness" is more suitable for Arabic. We decided later to remove it from our submission to avoid confusion to the reader. However, given that you have raised this very valid point, we will add it back. Please find a summary of our discussion of "dialectness" vs "formality" for Arabic below, which we will add to the camera-ready version of the paper.
>
> While we think the two concepts can be related, we will explain why we think the “level of dialectness” concept that we are introducing differs from “formality”, and is actually more suitable for Arabic:
>
> * During the literature review, we found that “formality” is a concept that has been widely studied, yet it does not generally have an agreed upon definition. Formality entails different stylistic dimensions such as serious-trivial, polite-casual and depends on the degree of shared knowledge [1]. (Pavlick et al, 2016) noticed that the annotator agreement scores are high for sentences at the extreme ends (i.e.: unambiguously formal or informal), but the agreement scores are lower otherwise [1].
>
> * In our paper, we define the “level of dialectness” as the degree by which a sentence diverges from the Standard Language. The “level of dialectness” is therefore topic-independent, and less ambiguous than “formality”.
>
> * To illustrate that the two concepts are different, and as demonstrated in Figure 4, we show that the level of dialectness of the political speeches of three Arab presidents is variable. Political speeches are typically considered to be formal, however, it is clear that the presidents might still use dialectal sentences in their speeches especially if they are spontaneously speaking, as in the speech visualized in Subfigure 4e. If you consider the following three sentences from this presidential press conference, which is a formal speech, you can see that the sentences have different levels of dialectness. However, if you define “formality” as a distinction between serious-trivial or polite-casual, then one can argue that all three sentences are highly serious and polite, which renders them all as being formal.
>
>
> | Predicted ALDi score  [0, 1]    | Sentence | Segment in Speech |
> | :---:        |    ----:   |          :---: |
> | 0 | واتفقنا على أن الوضع الحالي يفرض على كافة الفاعلين الدوليين التحلي بالمسؤولية لإيجاد حلول وآليات عملية تخفف من تداعيات الأزمة على الدول الأكثر تضررًا. | Main speech|
> | 0.41 | وأنا عايز أقول إن إحنا تحدث، يعني أنا تحدثت في هذا الأمر إن المطلوب التنسيق والتعاون بين كل دول العالم في ما يخص هذا الملف أثناء حديثي أو خطابي في مؤتمر جدة عن موضوع الطاقة تحديدًا. | Q&A - Gas |
> | 0.75 | وإحنا مش مهتمين بيه عشان أنتوا بتسألوا عنه.. مهم قوي إن إنتوا تعرفوا كدا.. إحنا مهتمين بيه عشان إحنا بنحترم شعوبنا، وبنحبها، ومش كلام، إحنا بنحترم شعوبنا زي ما إنتوا ما بتحترموا شعوبكم.. وبالتالي إحنا مش مهتمين عشان أنتوا بتسألونا عليه.. لأ.. ده مسؤوليتنا الأخلاقية والتاريخية والإنسانية تجاه شعوبنا. دي نقطة. | Q&A - Human Rights |
>
> * To the best of our knowledge, neither the "formality" nor the "level of dialectness" of **Arabic** text were studied before. Therefore,
> introducing the AOC-ALDi corpus that is derived from the AOC corpus would be a great addition to the Dialectal Arabic resources, and would hopefully motivate more research in this direction.
>
> * To conclude, we agree that there is a relation between formality and the level of dialectness, since Arabic speakers tend to use MSA in formal situations, and their regional dialects in informal ones. However, an Arabic speaker can still use MSA and speak informally, or use their dialect and speak formally. Therefore, we believe that they are two different concepts that are not interchangeable.
> * Finally, we thank you again for raising this important point, and we will add this discussion (with its corresponding references) to the camera-ready version of our paper.
>
> **Ans.**: For the question about the examples in Table 1, we believe that they are consistent with our definition of “level of dialectness” as a measure of the divergence from the Standard language. For the “Medium” level of dialectness example, the verb “بسطنا” has an MSA etymology [2], [3], which is generally used to mean expand something, yet can also mean “cheering someone”. For the “High” level of dialectness examples, the verbs “شهيصنا” and “نغنجنا” do not have an MSA root, which makes them more divergent from MSA, and thus are higher in dialectness.
>
>
> ***
>
> [1] Ellie Pavlick and Joel Tetreault. 2016. An Empirical Analysis of Formality in Online Communication. Transactions of the Association for Computational Linguistics, 4:61–74. (https://aclanthology.org/Q16-1005/)
>
>
> [2] https://en.wiktionary.org/wiki/%D8%A8%D8%B3%D8%B7
>
>
> [3] ​​https://www.almaany.com/ar/dict/ar-ar/%D8%A8%D8%B3%D8%B7/?c=%D8%A7%D9%84%D9%85%D8%B9%D8%AC%D9%85%20%D8%A7%D9%84%D9%88%D8%B3%D9%8A%D8%B7

---

### Meta-Review · Area_Chair_kfdi · 2023-09-17

**Recommendation:** 4

**Metareview:**

All reviewers agree that the paper contributes to an important research area, which is to add nuances of dialect usage by Arabic speakers in the existing work of Arabic Dialect Identification. While the idea of representing dialectness as a continuous variable isn't novel, the authors operationalize it and perform extensive annotations on the news articles and users' comments, where each sentence is annotated three times. Furthermore, one important highlight of the paper is to demonstrate the usefulness of the dataset for the sociolinguistic field, where the authors show that we can use models trained on their dataset to characterize the Arabic speaking stylistic choices (and even better than using vectors of scores from existing Dialect Identification models). During the post-rebuttal discussion, the authors also clarify the dimensional differences between dialectness and formality, and therefore this work contributes to the existing resources on formality and the binary representation of dialectness for Arabic dialect identification.

---

### Decision · Program_Chairs · 2023-10-07

**Decision:**

Accept-Main

**Comment:**

All reviewers agree that the paper contributes to an important research area, which is to add nuances of dialect usage by Arabic speakers in the existing work of Arabic Dialect Identification. While the idea of representing dialectness as a continuous variable isn't novel, the authors operationalize it and perform extensive annotations on the news articles and users' comments, where each sentence is annotated three times. Furthermore, one important highlight of the paper is to demonstrate the usefulness of the dataset for the sociolinguistic field, where the authors show that we can use models trained on their dataset to characterize the Arabic speaking stylistic choices (and even better than using vectors of scores from existing Dialect Identification models). During the post-rebuttal discussion, the authors also clarify the dimensional differences between dialectness and formality, and therefore this work contributes to the existing resources on formality and the binary representation of dialectness for Arabic dialect identification.